# Risk of Preeclampsia and Adverse Pregnancy Outcomes after Heterologous Egg Donation: Hypothesizing a Role for Kidney Function and Comorbidity

**DOI:** 10.3390/jcm8111806

**Published:** 2019-10-28

**Authors:** Federica Fassio, Rossella Attini, Bianca Masturzo, Benedetta Montersino, Antoine Chatrenet, Patrick Saulnier, Gianfranca Cabiddu, Alberto Revelli, Gianluca Gennarelli, Isabella Bianca Gazzani, Elisabetta Muccinelli, Claudio Plazzotta, Guido Menato, Giorgina Barbara Piccoli

**Affiliations:** 1Obstetrics, Department of Surgery, University of Torino, Torino 10100, Italy; federica.fassio@hotmail.it (F.F.); rossella.attini@gmail.com (R.A.); alberto.revelli@unito.it (A.R.); gianluca.gennarelli@unito.it (G.G.); isabella.gazzani@hotmail.com (I.B.G.); emuccinelli@cittadellasalute.to.it (E.M.); cplazzotta@cittadellasalute.to.it (C.P.); guido.menato@unito.it (G.M.); 2Nephrology, Centre Hospitalier Le Mans, Le Mans 72000, France; antoine.chatrenet@gmail.com; 3Laboratory of Statistics, University of Angers, Angers 49035, France; patrick.saulnier@chu-angers.fr; 4Nephrology, Brotzu Hospital, Cagliari 09134, Italy; gianfranca.cabiddu@tin.it; 5Department of Clinical and Biological Sciences, University of Torino, Torino 10100, Italy

**Keywords:** pregnancy, egg donation, preterm delivery, preeclampsia, kidney function

## Abstract

Background and objectives: Preeclampsia (PE) is a risk factor for kidney diseases; egg-donation (ED) increasingly used for overcoming fertility reduction, is a risk factor for PE. CKD is also a risk factor for PE. However, kidney function is not routinely assessed in ED pregnancies. Objective of the study is seeking to assess the importance of kidney function and maternal comorbidity in ED pregnancies. Design, setting, participants and measurements. Design: retrospective observational study from clinical charts. Setting: Sant’Anna Hospital, Turin, Italy (over 7000 deliveries per year). Selection: cases: 296 singleton pregnancies from ED (gestation > 24 weeks), who delivered January 2008–February 2019. Controls were selected from the TOrino Cagliari Observational Study (1407 low-risk singleton pregnancies 2009–2016). Measurements: Standard descriptive analysis. Logistic multiple regression analysis tested: PE; pregnancy-induced hypertension; preterm delivery; small for gestational age; explicatory variables: age; BMI; parity; comorbidity (kidney diseases; immunologic diseases; thyroid diseases; other). Delivery over time was analyzed according to Kaplan Meier; ROC (Relative Operating Characteristic) curves were tested for PE and pre-term delivery, employing serum creatinine and e-GFR as continuous variables. The analysis was performed with SPSS v.14.0 and MedCalc v.18. Results: In keeping with ED indications, maternal age was high (44 years). Comorbidity was common: at least one potential comorbid factor was found in about 40% of the cases (kidney disease: 3.7%, immunologic 6.4%, thyroid disease 18.9%, other-including hypertension, previous neoplasia and all other relevant diseases—10.8%). No difference in age, parity and BMI is observed in ED women with and without comorbidity. Patients with baseline renal disease or “other” comorbidity had a higher risk of developing PE or preterm delivery after ED. PE was recorded in 23% vs. 9%, OR: 2.513 (CI 1.066–5.923; *p* = 0.039); preterm delivery: 30.2% vs. 14%, OR 2.565 (CI: 1.198–5.488; *p* = 0.044). Limiting the analysis to 124 cases (41.9%) with available serum creatinine measurement, higher serum creatinine (dichotomised at the median: 0.67 mg/dL) was correlated with risk of PE (multivariate OR 17.277 (CI: 5.125–58.238)) and preterm delivery (multivariate OR 2.545 (CI: 1.100–5.892). Conclusions: Within the limits of a retrospective analysis, this study suggests that the risk of PE after ED is modulated by comorbidity. While the cause effect relationship is difficult to ascertain, the relationship between serum creatinine and outcomes suggests that more attention is needed to baseline kidney function and comorbidity.

## 1. Introduction

As egg donation has increasingly come to be used as an assisted fertilization technique, its main role has progressively shifted from making pregnancy possible in patients with primary or early ovarian failure, or with severe genetic diseases, to extending the age of pregnancy beyond the physiological limit of fertility [1,2,3].

Notwithstanding the indications, egg donation is associated with a higher risk of virtually all adverse pregnancy outcomes, including caesarean section, preterm delivery, small for gestational age babies and the hypertensive disorders of pregnancy [4,5,6,7,8,9]. A number of elements seem to contribute to increased risk: among them, the most commonly studied ones are advanced maternal age, multiple pregnancies, and the presence of specific diseases that cause infertility [10,11,12]. 

The risks of adverse pregnancy-related outcomes are at least in part linked to the procedure used. As a general rule, the more complex the procedure, the higher the risk; in vitro fertilisation is the most complex procedure, and the one with the highest risk [13,14]. Furthermore, immunologic factors probably play an important role, and the studies comparing pregnancy after homologous fertilisation and egg donation have reported an increased incidence of adverse events in the latter, in which the oocytes are 100% genetically different from the mother’s. Some studies suggest that HLA compatibility may be an important issue, leading some authors to consider egg donation as a form of “transplantation” without immunosuppression [15,16,17,18,19]. 

Even if the increase in comorbid conditions that goes along with maternal age is often mentioned as a possible cause for increased adverse pregnancy outcomes in egg-donor pregnancies, in which maternal age is usually higher, to date no study has focused on different comorbidity and, in this context, the “missing piece” is an analysis of kidney function, which is closely associated with pregnancy outcomes [20,21,22,23,24,25,26,27]. 

It was in this context that we undertook the present retrospective analysis. Our aim is to assess the prevalence and the role of the main comorbid conditions, with particular attention to the presence of autoimmune and kidney diseases, and to try to understand how they affect the main pregnancy outcomes in singleton pregnancies from heterologous egg donation. Furthermore, the relationship between kidney function data was explored in patients who had at least one available creatinine test during pregnancy or at delivery. 

## 2. Methods

### 2.1. Setting of Study

The study was performed at Ospedale Sant’Anna, a tertiary-care hospital, that is part of the Città della Salute e della Scienza in Torino, Italy; the low-risk controls were obtained from the archives of the Torino Cagliari Observational Study (TOCOS), dedicated to women with CKD, described elsewhere [21,24]. 

With an average of over 7000 deliveries per year, Sant’Anna Hospital is one of the largest European tertiary care obstetric facilities. 

Egg donation has only been available in Italy since 2014–2015 and even now, many restrictions apply to its use, in particular in high maternal age. Therefore, until 2015 patients who decided to have egg-donation pregnancies travelled outside the country, mainly to Spain, where these procedures are legal. As a consequence, pre-conception data are usually not available.

### 2.2. Selection of the Cases

#### 2.2.1. Egg-Donation

The selection of the cases of pregnancy in women undergoing egg-donation included all patients who delivered at the Sant’Anna Hospital between January 2008 and February 2019.

Pregnancies from egg-donation were identified in two main sources: the first was the hospital’s discharge database in which medical history, birth data and discharge letters are recorded. The search was carried out using the Italian keywords related to “egg-donation” in various combinations (e.g., “ovodonazione, ovodonazioni, ED eterologa, OD, fecondazione eterologa”). Secondly, we searched the computerised records of pregnant women referred to the Prenatal Screening Center at the Sant’ Anna Hospital, where information on type of conception is mandatorily recorded (egg-donation, other types of assisted fertilisation techniques, spontaneous pregnancy). 

#### 2.2.2. Low-Risk PREGNANCIES

Low-risk pregnancies were selected from the TOCOS archives (TOrino Cagliari Observational Study [21,24]). In this study, low-risk pregnancy is defined as a spontaneous singleton pregnancy occurring in a woman without pre-existing, systemic and localized diseases. The database presently gathers 1407 Low-risk singleton pregnancies observed in the two settings in the period 2009–2016.

### 2.3. Selection Criteria

From the available databases, we selected singleton pregnancies with a duration of more than 24 weeks, with complete data at delivery (gestational week, maternal age, birth centile). Abortions and therapeutic interruptions of pregnancy were excluded from the study.

### 2.4. Data Recorded

The following data regarding egg donation pregnancies were recorded: age, ethnicity, parity, baseline BMI; hypertension (pre-existing and gestational); diabetes (previous and gestational);diagnosis of PE; diagnosis of HELLP syndrome; gestational week of delivery, type of delivery; baby’s weight at birth, centile;need for maternal intensive care unit, need for neonatal intensive care unit, perinatal death, maternal death;

Whenever available, the following information was also included: previous PE, previous miscarriages, any other maternal disease; previous assisted fertilization; family history of CKD, cardiovascular disorders or preeclampsia.

Serum creatinine was recorded whenever available in the clinical charts or in the hospital laboratory records before and during hospitalization. Jaffé Method was employed up to 2017, when it was substituted by the enzymatic method. 

### 2.5. Statistical Analysis

Descriptive analysis was conducted as appropriate. The Shapiro Test was used to verify the normality distribution of the data. Median and min-max or range was used for non-parametric data, mean, and standard deviation for normal distribution. 

Statistical significance was assessed using ANOVA for parametric data and Kruskal-Wallis for non-parametric data, in compliance with standard indications for continuous variables. Dichotomous data are presented as risks, rates and proportions; in this case, the significance of the differences was analysed using the Chi-Square Test.

The significance of differences between groups was assessed using the *t*-test for normally distributed variables and the Wilcoxon Signed Rank Test for variables that were not normally distributed.

Logistic multiple regression analysis was performed for the covariates that emerged as significant in the descriptive analysis or were considered clinically relevant on the basis of the current understanding of the literature. Different outcomes were tested: preterm delivery (<37, <34 and <32 completed gestational weeks); small for gestational age baby (SGA, defined as < 5 and <10 centile, according the Italian birth reference charts, InES charts); preeclampsia (PE); pregnancy-induced hypertension (PIH); for this latter, only cases that were normotensive before pregnancy were considered. For the present study, we selected the outcomes of preterm delivery (<37 weeks), SGA < 10 centile and PE.

The outcomes were tested in the whole population of egg-donation pregnancies and in the subset of cases in which at least one serum creatinine assessment was available. 

The following dichotomized covariates were tested: age (dichotomized at the median); BMI (dichotomized at pre-gestational BMI of 30 kg/m^2^); parity (primiparous versus multiparous); presence of comorbidity (the following categories were assessed: thyroid diseases; immunologic diseases; kidney diseases; other relevant comorbidities, including diabetes, hypertension, history of neoplasia, genetic diseases; since immunologic and thyroid diseases needed to be in full remission or complete control, we tested both all comorbid conditions and kidney and other relevant diseases only). 

Analysis of delivery over time was conducted according to Kapan Meier (outcome: timing of delivery; stratification according to comorbidity, testing the two subsets with all comorbid conditions and kidney diseases plus other relevant comorbidities); comparison between curves was made using the Log-Rank Test. 

ROC curves were tested for the outcomes’ pre-term delivery and preeclampsia, employing serum creatinine and e-GFR as continuous variables.

The analysis was performed with SPSS software v.14.0 (IMB Corp., Chicago, IL, USA) and MedCalc v.18 (MedCalc Software bvba, Ostend, Belgium).

### 2.6. Ethical Issues

This is an observational retrospective study, encompassing an analysis of the clinical charts of women who delivered after egg donation in a single Italian center in the years 2017–2018; the aim of the study was to detect predisposing factors. All patients evaluated in the present study signed, at time of hospitalization, a standard consent form, authorizing anonymous data management and publication. The study was notified to the hospital’s ethics committee. 

## 3. Results

### 3.1. Characteristics of the Population

Table 1 summarizes the main characteristics of the study population. In keeping with what is known about egg-donor pregnancies, this cohort has a higher age at delivery, and a higher incidence of primiparity. About 40% of the cases display at least one comorbid condition, of which thyroid disorders are the ones most frequently reported. Immunologic disorders were also relatively frequent. Due to the fact that both needed to be in full remission-compensation as a requisite for egg donation, we further considered them separately, including or excluding them in the comorbidity analysis. 

As expected, pregnancy-related adverse outcomes have a significantly higher incidence in the cohort of patients fertilized through egg donation, compared with low-risk pregnancies. The differences remain highly significant when only “no- risk” patients are considered. 

It is important to emphasize that the preeclampsia rate is of 11.1% after egg donation as compared 1.7% in low risk controls, and that they have 3.7% vs. 0.5% of early preterm babies (<32 weeks, OR 7.7) while, quite surprisingly, SGA rate differs only when specific comorbidities are considered. Further, it is of note that women who underwent egg donation have a significantly higher incidence of LGA babies (12.5% vs. 7.6% in controls).

No difference in age, parity and BMI is observed in women with and without comorbid conditions who had been fertilized through egg donation. Conversely, a trend towards an increase in adverse pregnancy-related events is observed in patients with comorbidity who had received egg donation (all conditions together) and the difference reaches statistical significance for gestational hypertension, centiles and incidence of small for gestational age babies, which is roughly double in “at risk” mothers.

### 3.2. Relationship between Comorbidity and Outcomes

The most commonly reported comorbid condition found was thyroid disease. Chronic kidney disease was present in 11 patients (3.7%). The diagnoses were kidney malformations (3); glomerular diseases (2); lupus nephropathy (3); kidney stones (2); diabetic nephropathy (1). One patient, previously on dialysis for lupus nephropathy, was a kidney graft recipient at the time of egg donation. She was the only one who was followed up in a nephrology-obstetric outpatient facility.

Other comorbid conditions included: chronic hypertension (10 cases without CKD); cardiopathy (4); previous neoplasia treated by chemotherapy (5); fragile X or trisomy X syndromes (3); MTHFR homozygous mutation (5); other causes (5).

The different “risk factors” do not appear to have the same weight. Thyroid diseases and other autoimmune diseases, do not appear to affect the risk of developing preeclampsia and of having a small for gestational age baby, or preterm delivery, as happens with kidney diseases and other comorbid condition (Table 2). The incidence of caesarean section is not higher in this subset, suggesting the presence of “late preterm” deliveries. 

The prevalence of small for gestational age babies was apparently not affected by the presence of comorbidity (considered all together), but was twice as frequent after egg donation than in the control population (Table 1). More in detail, in the subset of women who underwent egg donation, the presence of renal, immunologic or other comorbidity was associated with a higher prevalence of small for gestational age babies, with respect to no risk cases or to individuals with a history of thyroid diseases (Table 2).

### 3.3. Characteristics of the Patients with at least One Measure of Serum Creatinine

To further analyse the relationship between kidney function and maternal-foetal outcomes, we analysed the kidney function data available during pregnancy. Due the particular characteristics of the study population (delivery in a public hospital, but assisted fertilisation procedure often performed in a private structure, frequently outside Italy), creatinine assessment, which is not a part of routine assessments in pregnancy, was available in only 42% of the cases, generally only at delivery. 

In this context, creatinine was more frequently assessed in complicated pregnancies, mainly in the context of preeclampsia and preterm delivery (Table 3). This is also witnessed by the association of creatinine availability with preeclampsia (24.2% vs. 1.7%) and with small for gestational babies (12.9% vs. 6.4%).

### 3.4. Relationship between Serum Creatinine and eGFR and Materno-Foetal Outcomes

In the context of a “negative selection” of patients with egg-donor pregnancies, the patients with higher creatinine or lower glomerular filtration rates were not different from those with lower creatinine (or higher e-GFR) in terms of age, presence of all comorbidities, parity and BMI. 

A higher creatinine level was significantly associated with preeclampsia, gestational hypertension and pre-term delivery; conversely, the incidence of small for gestational age babies was not different across eGFR groups (Table 4).

ROC curves support the significant relationship between serum creatinine and e-GFR, analysed as continuous variables, preterm delivery and preeclampsia. The area under the curve is approximately 70% in both cases (Figure 1). 

The Youden index for the relationship between e-GFR and preterm delivery or pre-eclampsia is at ≤96 mL/min, and is at >0.64 mg/dL for preterm, and >0.63 mg/dL for the relationship between pre-eclampsia and serum creatinine. Interestingly, both indexes were in proximity of median values, respectively 0.67 mg/dL for serum creatinine and 103 mL/min for e-GFR (Table 2).

### 3.5. Multivariate Analysis

The multivariate analysis, shown in Table 5 and Table 6, suggests that the presence of pre-pregnancy comorbidity (renal diseases and other relevant disorders) is associated with a roughly double risk for preeclampsia and preterm delivery (both statistically significant: Preeclampsia: OR: 2.53; delivery <37 weeks: 2.56), while, even if only comorbidity is retained in the last step, statistical significance is not reached in the case of small for gestational age babies (OR 2.43). 

In the cohort in which serum creatinine assessment was available, the importance of this covariate is higher than that of comorbidity, and statistical significance is once more reached for preeclampsia and preterm delivery (OR 17.27 for preeclampsia) (Table 6).

In both cohorts (all cases and only patients with one assessment of serum creatinine), none of the covariates was significantly associated with early preterm delivery, defined as <34 or <32 gestational weeks.

### 3.6. Analysis of Timing of Delivery

Figure 2 shows the Kaplan Maier analysis (outcome: timing of delivery) sorting patients according to the presence of baseline comorbidity. The presence of baseline comorbidity is significant and remains so in the subset of patients with serum creatinine assessment.

The presence of a higher serum creatinine or a lower eGFR is likewise significantly correlated with timing of delivery (Figure 3).

## 4. Discussion

Egg donation is increasingly being used as a fertilization procedure. It has proved to be effective in “extreme” cases of infertility or early ovarian failure, but involves a higher risk for adverse pregnancy outcomes, including pre-term delivery, preeclampsia and related disorders and giving birth to a child that is small for gestational age [7,8,9,10,11,12,13,14,15,16,17]. 

While several studies have addressed obstetric and procedure-related risk factors, as well as at maternal age, the role of immunological diversity and the role of baseline maternal comorbidity have not yet been fully addressed. 

Various maternal conditions are considered as potentially affecting maternal-foetal outcomes in spontaneous pregnancies; however, less is known about their effect on pregnancies obtained by assisted fertilization techniques. Seeking to determine whether the “high risk” procedure of egg donation offsets the effect of baseline comorbidity, we undertook the present retrospective analysis, testing the effect of different classes of comorbidity on the outcomes of pregnancies after egg donation. Furthermore, within the limits later discussed in detail, we tried to explore the relationship of kidney function in this context, taking also into account the relationship between preeclampsia and long-term kidney health [28,29,30,31,32].

As expected in a relatively old cohort of pregnant women, comorbidity was relatively common. 3.7% of the women were affected by a chronic kidney disease, which was however recognized as a high risk condition in only one case (a kidney transplant recipient); 6.4% had a history of an autoimmune disorder other than lupus and not a cause of kidney disease (all of them in remission); thyroid diseases, once more in metabolic balance, were frequently reported (18.9%), probably also because of the attention paid to these conditions in women with reduced fertility, while other relevant comorbidities, including baseline hypertension and diabetes or previous neoplasia, were reported in 10.8% of the pregnancies (Table 1 and Table 2).

The different comorbidities do not appear to play the same role: a history of thyroid disorders and of immunologic diseases was not associated with an increase in adverse pregnancy outcomes. Conversely, kidney diseases and other relevant comorbidities were associated with approximately twice as many preterm deliveries and cases of preeclampsia, while the increase in small for gestational age babies did not reach statistical significance in univariate and multivariate analysis (Table 2 and Table 5). 

The presence of pre-conception comorbidity significantly affects the kinetics of delivery, as was shown using the Kaplan Meier curves (Figure 2).

Given the importance of the kidney function in determining pregnancy outcomes, we further tried to analyze the subset of patients with available kidney functional data [20,21,22,23,24,26]. This subset accounts for only about 40% of all cases; selection was not random, since serum creatinine was more commonly available in women with baseline comorbidity, preeclampsia or preterm delivery, while age, parity and BMI were equally distributed (Table 3). This finding stresses the importance of higher awareness of the role of kidney function and of systematic assessment of serum creatinine in all pregnancies (and even more so in pregnancies with potential risk factors, including egg donation). Such a policy may also allow preemptive interventions, such as aspirin prophylaxis, and strict monitoring in patients with reduced kidney function.

Within the limits of a negative selection of cases, preeclampsia and pre-term delivery were significantly associated with higher creatinine levels, and consequently lower e-GFR levels. Interestingly, the association is present with a cut-point that is still in the normal serum creatinine range, defined using Youden’s index (Figure 1, Table 4 and Table 6). 

The close association between pregnancy outcomes and kidney function has to be viewed with caution. First of all the selection is probably biased, and a higher number of cases with suboptimal kidney function may be present in the subset of cases with available data; the low number of data did not allow correlating severity of preeclampsia or early delivery with creatinine level. However, the relationship between complications and higher creatinine level suggests, in keeping with the data in the literature, that adding more non-complicated pregnancies would strengthen the results. Furthermore, available kidney functional data were mainly measured shortly before delivery, and, in this context, it is not possible to conclude that there is a cause effect relationship between relative kidney function reduction and complications. However, the association is close enough to suggest that kidney function should be monitored from preconception counseling throughout pregnancy in all women undergoing egg donation. 

This study has the strong point of novelty, being the first one addressed at establishing the role of preconception comorbidity, in a population of women with pregnancies obtained via heterologous egg donation. However, it suffers from relevant methodological limitations: the assessment of comorbidity was retrospective and based upon clinical charts, and it is possible that the attention to classic risk factors for infertility (such as thyroid or immunologic diseases) was higher in comparison with other conditions. This is also suggested by the fact that only one of the patients with a clinical history of chronic kidney disease had been identified and referred to the multidisciplinary follow-up facility. In this regard, the experience discussed in this study supports the need for an in-depth analysis of comorbidity in women undergoing egg donation, and hypothesizes that timely referral to specialized care could reduce the more than two-fold increase in preeclampsia and preterm delivery we observed. From a research standpoint, our data make it possible to define a prospective data set that should be gathered for future analyses. 

Within these limits, and in the wait for a prospective study, we feel that our analysis is worth being shared, as it draws attention to two crucial elements, baseline comorbidity and kidney function, whose influence is not offset by the effect of the complex fertilization procedure, and which should be acknowledged and systematically controlled as potential targets for reducing the incidence of adverse maternal-fetal outcomes in pregnancies from heterologous egg donation. 

## 5. Conclusions

Risks associated with pregnancy after egg donation is significantly influenced by baseline comorbidity. Furthermore, within the limit of incomplete data and of a negative selection of the cases with available data, pregnancy outcomes are associated with higher creatinine levels, even within the normal serum-creatinine range. While further studies are needed to determine the cause and effect relationship of this association, the association is close enough to suggest that kidney function should be monitored from preconception counseling, throughout pregnancy in all women, and in particular, in high risk situations, including women undergoing egg donation. 

## Figures and Tables

**Figure 1 jcm-08-01806-f001:**
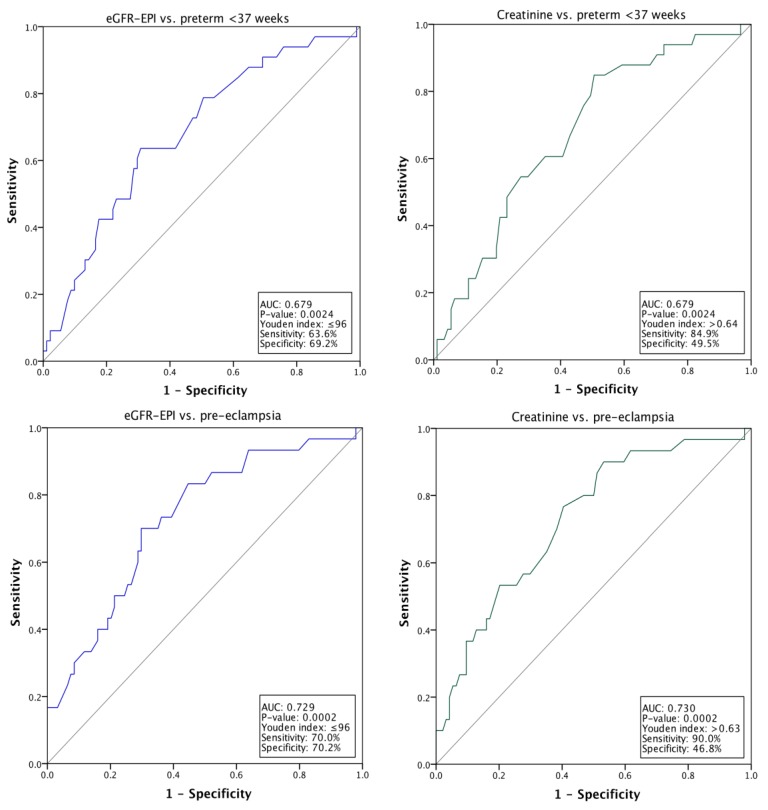
ROC (Relative Operating Characteristic) Curves: relationship between Preterm delivery or Preeclampsia and serum creatinine or eGFR.

**Figure 2 jcm-08-01806-f002:**
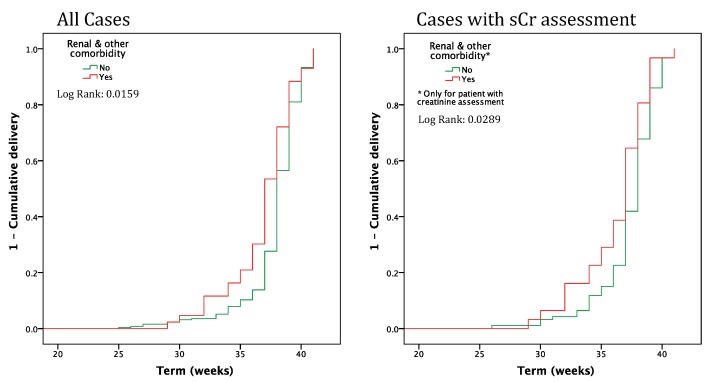
Timing of delivery in pregnancies from egg donation, with and without “renal or other” comorbidity (all cases and cases with serum creatinine assessment). sCr: Serum creatinine.

**Figure 3 jcm-08-01806-f003:**
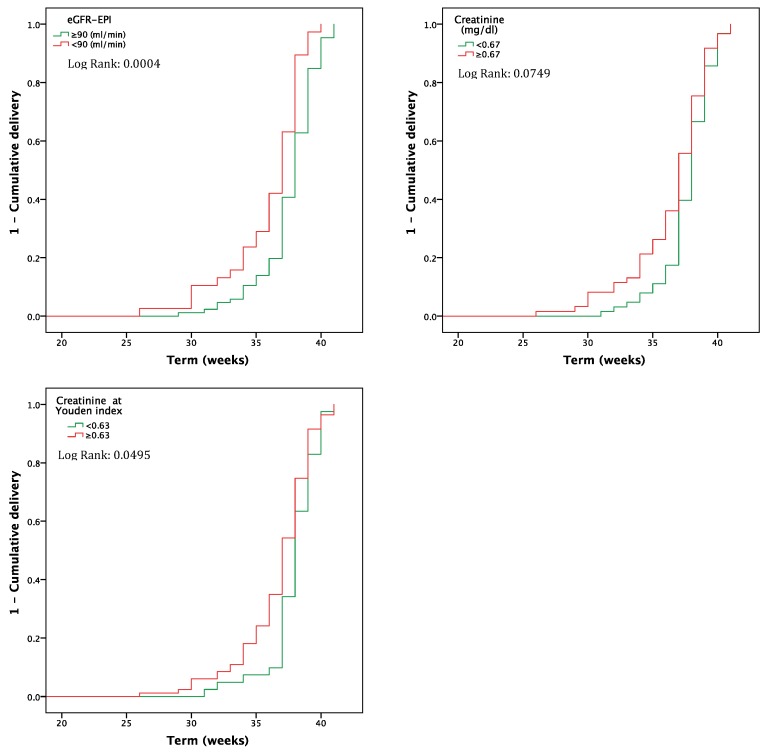
Timing of delivery in pregnancies from egg donation, according to serum creatinine (dichotomised at the median) and to e-GFR.

**Table 1 jcm-08-01806-t001:** Baseline and main delivery data in pregnancies from egg donation and in controls.

	Whole Population			Egg Donation		
	Egg donation (*n* = 296)	Controls (*n* = 1407)	*p*-Values	No Conorbidity (*n* = 178)	Comorbidity (*n* = 118)	*p*-Values
Age (years), median (min–max)	44 (31–56)	31 (15–49)	**<0.0001**	44 (31–56)	43 (33–52)	0.3797
Parity, *n*. first (%)	259 (87.5%)	803 (57.1%)	**<0.0001**	151 (84.8%)	108 (91.5%)	0.0887
BMI (kg·m^−2^), median (min–max)	22.7 (15.6–35.5)	22.4 (15.0–39.7)	0.9102	22.7 (16.8–33.7)	23.6 (15.6–35.5)	0.9598
Hypertension during pregnancy, *n* (%) *	74 (25.0%)	-	**-**	31 (17.4%)	43 (36.4%)	**0.0002**
Gestational hypertension, *n* (%)	60 (20.3%)	-	**-**	31 (17.4%)	29 (24.6%)	0.1342
Preeclampsia, *n* (%)	33 (11.1%)	24 (1.7%)	**<0.0001**	16 (9.0%)	17 (14.4%)	0.1477
Centiles at birth, median (min–max)	55 (0–100)	45 (0–100)	**0.0051**	60 (1–100)	48 (0–100)	**0.0071**
Creatinine (mg·dL^−1^), median (min–max)/*n*	0.67 (0.26–1.94) / 124	-	**-**	0.67 (0.26–1.94) /61	0.68 (0.46–1.70) /63	0.3197
eGFR-EPI (mL·min^−1^), median (min–max)	103 (31–151)	-	**-**	103 (33–151) / 61	103 (31–129) / 63	0.7929
GFR < 90 (mL·min^−1^), *n* (%)	38 (30.6%)	-	**-**	18 (29.5%) /61	20 (31.7%) /63	0.7878
Caesarean section, *n* (%)	210 (70.9%)	265 (23.6%)	**<0.0001**	128 (71.9%)	82 (69.5%)	0.6542
Size for GA Small, *n* (%) Adequate, *n* (%) Large, *n* (%)	27 (9.1%) 232 (78.4%) 37 (12.5%)	142 (10.1%) 1154 (82.0%) 107 (7.6%)	**0.0230**	12 (6.7%) 138 (77.5%) 28 (15.8%)	15 (12.7%) 94 (79.7%) 9 (7.6%)	**0.0380**
Pre-term <37 weeks, *n* (%) <34 weeks, *n* (%) <32 weeks, *n* (%)	48 (16.2%) 18 (6.1%) 11 (3.7%)	88 (6.3%) 13 (0.9%) 7 (0.5%)	** ** **<0.0001** **<0.0001** **<0.0001**	24 (13.5%) 10 (12.8%) 7 (3.9%)	24 (20.3%) 8 (6.8%) 4 (3.4%)	0.1178 0.6827 0.8093

eGFR EPI: glomerular filtration rate calculated with the CKD-EPI formula; BMI: body mass index; GA: gestational age. HT Hypertension. * Hypertension in pregnancy: all cases regardless of previous hypertension. Gestational hypertension is hypertension during pregnancy in previously normotensive patients. Significant results in bold.

**Table 2 jcm-08-01806-t002:** Baseline and main delivery data in pregnancies from egg donation according to the comorbid conditions considered.

	According to the Main Conditions					*p*-Value	According to Groups		*p*-Value
	No Risk	Thyroid	Auto-Immune	Renal	Other		Autoimmune and Thyroid	Renal and Other	
*n* (%)	178 (60.1%)	56 (18.9%)	19 (6.4%)	11 (3.7%)	32 (10.8%)		75 (25.3%)	43 (14.5%)	
Age (years), median (min–max)	44 (31–56)	43 (33–50)	44 (34–51)	43 (34–47)	45 (33–52)	0.5956	43 (33–51)	43 (33–52)	0.7599
Parity, *n*. first (%)	151 (84.8%)	48 (85.7%)	19 (100%)	11 (100%)	30 (93.8%)	0.1497	67 (89.3%)	41 (95.3%)	0.2608
BMI (kg·m^−2^), median (min–max)	22.7 (16.8–33.7)	21.5 (17.6–33.3)	21.8 (19.4–34.4)	24.6 (18.7–34.0)	25.2 (15.6–35.5)	0.2246	21.6 (17.6–34.4)	24.9 (15.6–35.5)	**0.0267**
Hypertension during pregnancy, *n* (%) *	31 (17.4%)	18 (32.1%)	5 (26.3%)	7 (63.6%)	13 (40.6%)	**0.0005**	23 (30.7%)	20 (40.6%)	**0.0865**
Gestational hypertension, *n* (%)	31 (17.4%)	18 (32.1%)	4 (21.1%)	5 (45.5%)	2 (6.2%)	**0.0073**	22 (29.3%)	7 (16.3%)	0.1145
Pre-eclampsia, *n* (%)	16 (9.0%)	6 (10.7%)	1 (5.3%)	4 (36.4%)	6 (18.8%)	**0.0336**	7 (9.3%)	10 (23.3%)	**0.0390**
Centiles at birth, median (min–max)	60 (1–100)	48 (1–99)	54 (4–100)	59 (1–95)	44 (0–90)	0.0949	48 (1–100)	46 (0–95)	0.4862
Caesarean section, *n* (%)	128 (71.9%)	39 (69.9%)	17 (89.5%)	8 (72.7%)	18 (56.2%)	**0.0177**	56 (74.7%)	26 (60.5%)	**0.0072**
Size for GA Small, *n* (%) Adeq, *n* (%) Large, *n* (%)	12 (6.7%) 138 (77.5%) 28 (15.8%)	4 (7.1%) 47 (83.9%) 5 (8.9%)	3 (15.8%) 13 (68.4%) 3 (15.8%)	3 (27.3%) 7 (63.6%) 1 (9.0%)	5 (15.6%) 27 (84.4%) 0 (0%)	0.0595 **	7 (9.3%) 60 (80.0%) 8 (10.7%)	8 (18.6%) 34 (79.1%) 1 (2.3%)	0.1139
Pre-term <37, *n* (%) <34, *n* (%) <32, *n* (%)	24 (13.5%) 10 (12.8%) 7 (3.9%)	9 (16.1%) 3 (5.4%) 2 (3.5%)	2 (10.5%) 0 0	3 (27.3%) 1 (9.0%) 0	10 (31.3%) 4 (12.5%) 2 (6.3%)	0.1014 0.4293 0.7800	11 (14.7%) 3 (4.0%) 2 (2.7%)	13 (30.2%) 5 (11.6%) 2 (4.7%)	**0.0441** 0.1142 0.5681

eGFR EPI: glomerular filtration rate calculated with the EPI formula; BMI: body mass index; GA: gestational age; PE: preeclampsia; Adeq. Adequate. HT Hypertension * Hypertension in pregnancy: all cases regardless of previous hypertension. Gestational hypertension is hypertension during pregnancy in previously normotensive patients. ** Chi Square test for the difference in the distribution. Significant results in bold.

**Table 3 jcm-08-01806-t003:** Comparison of subsets of egg donation pregnancies with or without an available creatinine test.

	All	sCr Available	sCr not Available	*p*-Values
*n* (%)	296 (100%)	124 (41.9%)	172 (58.1%)	
Age (years), median (min–max)	44 (31–56)	44 (31–56)	43 (33–52)	0.0813
Parity, *n*. first (%)	259 (87.5%)	116 (93.6%)	143 (83.1%)	**0.0077**
BMI (kg·m^−2^), median (min–max)	22.7 (15.6–35.5)	22.6 (15.6–35.5)	22.6 (16.8–33.7)	0.4934
Presence of comorbidity Autoimmune, *n* (%) Thyroid, *n* (%) Renal, *n* (%) Other, *n* (%) None, *n* (%)	19 (6.4%) 56 (18.9%) 11 (3.7%) 32 (10.8%) 178 (60.1%)	11 (8.9%) 21 (16.9%) 10 (8.1%) 21 (16.9%) 61 (49.2%)	8 (4.7%) 35 (20.3%) 1 (0.6%) 11 (6.4%) 117 (68.0%)	**0.0001 ****
Centile at birth, median (min–max)	55 (0–100)	54 (0–100)	58 (1–100)	0.0908
Pre-eclampsia, *n* (%)	33 (11.1%)	30 (24.2%)	3 (1.7%)	**<0.0001**
Hypertension during pregnancy, *n* (%) *	74 (25.0%)	64 (51.6%)	10 (5.8%)	**<0.0001**
Gestational hypertension, *n* (%)	60 (20.3%)	51 (41.1%)	9 (5.2%)	**<0.0001**
Caesarean section, *n* (%)	210 (70.9%)	98 (79.0%)	112 (65.1%)	**0.0094**
Size for GA Small, *n* (%) Adeq, *n* (%) Large, *n* (%)	27 (9.1%) 232 (78.4%) 37 (12.5%)	16 (12.9%) 97 (78.2%) 11 (8.9%)	11 (6.4%) 135 (78.5%) 26 (15.1%)	0.0610 **
Pre-term <37, *n* (%) <34, *n* (%) <32, *n* (%)	48 (16.2%) 18 (6.1%) 11 (3.7%)	33 (26.6%) 11 (8.9%) 6 (4.8%)	15 (8.7%) 7 (4.1%) 5 (2.9%)	**<0.0001** 0.0887 0.3868

eGFR EPI: glomerular filtration rate calculated with the EPI formula; BMI: body mass index; GA: gestational age; PE: preeclampsia; Adeq. Adequate; sCr: Serum creatinine. HT Hypertension. * Hypertension in pregnancy: all cases regardless of previous hypertension. Gestational hypertension is hypertension during pregnancy in previously normotensive patients. ** Chi Square test for the difference in the distribution. Significant results in bold.

**Table 4 jcm-08-01806-t004:** Comparison of egg donation pregnancies according to serum creatinine and eGFR.

	According to Creatinine (mg/dL)		*p*-Values	According to eGFR (mL/min)		*p*-Values
	sCr ≥ 0.67	sCr < 0.67		GFR < 90	GFR ≥ 90	
*n* (%)	68 (54.8%)	56 (45.2%)		38 (30.6%)	86 (69.4%)	
Age, median (min–max)	44 (32–55)	44 (31–56)	0.8363	45 (38–55)	44 (31–56)	0.1993
Parity, *n*. first (%)	64 (94.1%)	52 (92.9%)	0.7770	36 (94.7%)	80 (93.0%)	0.7214
BMI (kg·m^−2^), median (min–max)	22.0 (15.6–35.5)	22.8 (17.1–34.4)	0.8889	22.3 (15.6–34.0)	22.8 (17.1–35.5)	0.8816
No comorbidity, *n* (%)	31 (45.6%)	30 (53.6%)	0.3781	18 (47.4%)	60 (69.8%)	0.7878
Centiles at birth, median (min–max)	54 (0–98)	55 (1–100)	0.4483	40 (0–98)	58 (0–100)	0.1172
Pre-eclampsia, *n* (%)	24 (35.3%)	6 (10.7%)	**0.0015**	15 (39.5%)	15 (17.4%)	**0.0085**
Hypertension during pregnancy, *n* (%) *	43 (63.2%)	21 (37.5%)	**0.0045**	29 (76.3%)	35 (40.7%)	**0.0003**
Gestational hypertension, *n* (%)	31 (45.6%)	20 (35.7%)	0.2681	19 (50.0%)	32 (37.2%)	0.1838
Creatinine (mg·dL^−1^), median (min max)	0.75 (0.67–1.94)	0.57 (0.26–0.66)	**<0.0001**	0.86 (0.69–1.94)	0.63 (0.26–0.86)	**<0.0001**
GFR-EPI (mL·min^−1^), median (min-max)	87 (31–114)	114 (93–151)	**<0.0001**	78 (31–89)	108 (90–151)	**<0.0001**
Caesarean section, *n* (%)	53 (77.9%)	45 (80.4%)	0.7432	32 (84.2%)	66 (76.7%)	0.3484
Size for GA Small, *n* (%) Adeq, *n* (%) Large, *n* (%)	9 (13.2%) 55 (80.9%) 4 (5.9%)	7 (12.5%) 42 (75.0%) 7 (12.5%)	0.4350	6 (15.8%) 29 (76.3%) 3 (7.9%)	10 (11.6%) 68 (79.1%) 8 (9.3%)	0.8033 **
Pre-term <37, *n* (%) <34, *n* (%) <32, *n* (%)	25 (36.8%) 9 (13.2%) 5 (7.4%)	8 (14.3%) 2 (3.6%) 1 (1.8%)	**0.0050** 0.0607 0.1522	16 (42.1%) 6 (15.8%) 4 (10.5%)	17 (19.8%) 5 (5.8%) 2 (2.3%)	**0.0098** 0.0728 0.0507

eGFR EPI: glomerular filtration rate calculated with the EPI formula; BMI: body mass index; GA: gestational age; PE: preeclampsia; Adeq. Adequate. HT Hypertension. * Hypertension in pregnancy: all cases regardless of previous hypertension. Gestational hypertension is hypertension during pregnancy in previously normotensive patients. ** Chi Square test for the difference in the distribution. Significant results in bold.

**Table 5 jcm-08-01806-t005:** Multivariate analysis of different outcomes: all cases.

**Preeclampsia**		***p*-Values**	**OR**	**CI 95% OR**	
				**Lower**	**Higher**
First step	Age (dichotomised at the median: 44y)	0.326	1.456	0.688	3.082
	BMI (dichotomised at 30 kg/m^2^)	0.306	1.781	0.589	5.381
	Parity (1st vs. other)	0.165	0.236	0.031	1.807
	Comorbidity (renal and other)	0.056	2.355	0.979	5.667
Last step	Parity	0.152	0.227	0.030	1.725
	**Comorbidity (renal and other)**	**0.035**	**2.513**	**1.066**	**5.923**
**Delivery < 37 Weeks**		***p*-Values**	**OR**	**CI 95% OR**	
				**Lower**	**Higher**
First step	Age (dichotomised at the median)	0.910	0.963	0.504	1.840
	BMI (dichotomised at 30 kg/m^2^)	0.694	1.244	0.418	3.700
	Parity (1st vs other)	0.914	0.946	0.342	2.615
	Comorbidity (renal and other)	0.023	2.477	1.136	5.398
Last step	**Comorbidity (renal and other)**	**0.015**	**2.565**	**1.198**	**5.488**
**Small for Gestational Age**		***p*-Values**	**OR**	**CI 95% OR**	
				**Lower**	**Higher**
First step	Age (dichotomised at the median)	0.757	0.876	0.380	2.020
	BMI (dichotomised at 30 kg/m^2^)	0.291	0.325	0.040	2.617
	Parity (1st vs other)	0.484	0.586	0.131	2.622
	Comorbidity (renal and other)	0.048	2.634	1.011	6.866
Last step	Comorbidity (renal and other)	0.060	2.453	0.961	6.257

Legend: OR: odds ratio; CI: confidence intervals. BMI: body mass index. First and last step shown in tables (complete analysis available in Appendix A). Significant results in bold.

**Table 6 jcm-08-01806-t006:** Multivariate analysis of different outcomes: cases with creatinine assessment.

**Preeclampsia**		***p*-Values**	**OR**	**CI 95% OR**	
				**Lower**	**Higher**
First step	Age (dichotomised at the median)	0.472	1.342	0.603	2.986
	BMI (dichotomised at 30 kg/m^2^)	0.339	1.478	0.664	3.291
	Parity (1st vs. other)	0.312	0.340	0.042	2.756
	Comorbidity (renal and other)	0.880	1.063	0.480	2.357
	sCreat. (dichotomised at the median)	<0.001	15.809	4.631	53.968
**Last step**	**sCreat. (dichotomised at the median)**	**<0.001**	**17.277**	**5.125**	**58.238**
**Delivery <37 weeks**		***p*-Values**	**OR**	**CI 95% OR**	
				**Lower**	**Higher**
First step	Age (dichotomised at the median)	0.864	1.075	0.467	2.474
	BMI (dichotomised at 30 kg/m^2^)	0.616	0.689	0.161	2.955
	Parity (1st vs. other)	0.972	1.031	0.186	5.711
	Comorbidity (renal and other)	0.346	1.596	0.603	4.224
	sCreat. (dichotomised at the median)	0.062	2.315	0.958	5.593
**Last step**	**sCreat. (dichotomised at the median)**	**0.029**	**2.545**	**1.100**	**5.892**
**Small for Gestational Age**		***p*-Values**	**OR**	**CI 95% OR**	
				**Lower**	**Higher**
First step	Age (dichotomised at the median)	0.550	1.402	0.463	4.243
	BMI (dichotomised at 30 kg/m^2^)	0.585	0.543	0.061	4.865
	Parity (1st vs. other)	0.779	1.377	0.148	12.830
	Comorbidity (renal and other)	0.082	3.038	0.867	10.644
	sCreat. (dichotomised at the median)	0.535	0.687	0.210	2.251
Last step	Comorbidity (renal and other)	0.141	2.333	0.755	7.210

Legend: OR: odds ratio; CI: confidence intervals. BMI: body mass index. Note: Age median: 44 years; sCreat. Serum creatinine: median: 0.67 mg/dL; First and last step shown in tables (complete analysis available in Appendix A). Significant results in bold.

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
