# Peer review of "Risk of Preeclampsia and Adverse Pregnancy Outcomes after Heterologous Egg Donation: Hypothesizing a Role for Kidney Function and Comorbidity"

_jcm, 2019, doi:10.3390/jcm8111806_

Round 1

Reviewer 1 Report

The authors have revised the manuscript satisfactorily. There are no further comments.

Reviewer 2 Report

After modifications, this paper is perfect

This manuscript is a resubmission of an earlier submission. The following is a list of the peer review reports and author responses from that submission.

Round 1

Reviewer 1 Report

This an interesting study investigating kidney function, which is closely associated with normal and pathological pregnancy outcomes. 

It's understandable that serum creatinine data was not available for all the samples making sample size small even then significant conclusions were made.

As mentioned on Page 6, line 206-8, "creatinine assessment, which is not a part of routine assessments in pregnancy, was available in only 42% of the cases; generally only at delivery". This suggests that the conclusions must be made with caution due to the following reasons-

1- The onset of PE is usually early around 20 weeks of pregnancy and there is no creatinine data available for this or earlier time point. 

2- Is there any correlation between the severity of PE and creatinine levels measured during later phases of pregnancy?

2- It's important to mention What methods were used to determine creatinine levels and if there were similar or different sensitivities. 

Author Response

Thank you for reviewing our paper, for your kind words and for your keen comments.

Please find the point by point answers as follows: (in orange in the text)

Comment:

This an interesting study investigating kidney function, which is closely associated with normal and pathological pregnancy outcomes. It's understandable that serum creatinine data was not available for all the samples making sample size small even then significant conclusions were made.

answer: thanks, indeed, one of the pragmatic ideas was to support the systematic assessment of the kidney function in all pregnancies, starting from the high risk ones, such as those derived from egg donation. 

As mentioned on Page 6, line 206-8, "creatinine assessment, which is not a part of routine assessments in pregnancy, was available in only 42% of the cases; generally only at delivery". This suggests that the conclusions must be made with caution due to the following reasons-

1- The onset of PE is usually early around 20 weeks of pregnancy and there is no creatinine data available for this or earlier time point. 

2- Is there any correlation between the severity of PE and creatinine levels measured during later phases of pregnancy?

Pity, this was not possible; 

we added the following remark: the low number of data did not allow correlating severity of preeclampsia or early delivery with creatinine level.

2- It's important to mention What methods were used to determine creatinine levels and if there were similar or different sensitivities. 

We added the information in the methods: thanks for this precision. 

Jaffé Method was employed up to 2017, when it was substituted by the enzymatic method.  

Answer:

we smoothed the conclusion as follows:

Risks associated with pregnancy after egg donation are significantly influenced by baseline comorbidity. Furthermore, within the limit of incomplete data and of a negative selection of the cases with available data, pregnancy outcomes are associated with higher creatinine levels, even within the normal serum-creatinine range. While further studies are needed to determine the cause and effect relationship of this association, the association is close enough to suggest that kidney function should be monitored from preconception counseling, throughout pregnancy in all women undergoing egg donation.

and in the discussion we added:

This finding stresses the importance of higher awareness of the role of kidney function and of systematic assessment of serum creatinine in all pregnancy (and even more so in pregnancies with potential risk factors, including egg donation).

thanks for your comments, and for the suggestions,

the authors 

Reviewer 2 Report

Risk of preeclampsia and adverse pregnancy outcomes after heterologous egg donation. Hypothesizing a role for kidney function and comorbidity. Fassio et al

The subject is interesting for teams who have to manage women with egg donations (ED). First, this cohort is impressive: 296 pre or menopaused  women (44 years) in 11 years (ap. 27 per year).

The paper is well written, the introduction, the abstract, the discussion are well conducted with good references. The methodology and calculations are good. The authors explain well the limitations of the study and that, however, they may highlight their findings on kidney function, and that screening of creatinine may be of importance in these women (and put them under low-dose aspirin if pathological for example?).

In Tables 1 to 4. I would recommend to skip Adequate and LGA in these 4 tables and keep only the SGA in “size for GA”. First, because SGA is the more important, and, second, because in the p value column, Table 1, we obtain a global 0.02 (“size for GA”), while SGA is 9.1% vs 10.1%, adequate 78% vs 82%. The difference indeed comes from the LGA (!) 12.5% (ED) vs 7.6% ctrls, OR 1.7 [1.2-2.6], p= 0.002.

The Tables give a lot of informations, and there is no place to put the odds ratios in additional columns, however, I would recommend to write down some of them when they are important in the results:

In comments of Table 1, paragraph 3.1.

It is important to emphasize first that the preeclampsia rate is of 11.1% in ED women as compared with controls of the whole population (1.7%). OR 7.2 [4.2-12.5], and that they have 3.7% vs 0.5% of premature < 32 weeks OR 7.7 [3.0-20], while, surprisingly, they have the same rate of SGA. This is no more the case when we consider ED’s: morbid vs no-morbid SGA 12.7% vs 6.7% OR 2.01 [0.90-4.6], p= 0.04. Further, it is of note that ED women have significantly more LGA than controls 12.5% (ED) vs 7.6% ctrls, OR 1.7 [1.2-2.6], p= 0.002 (not shown in the Table).

Table 2. Table 2 is on page 9 (after Table 3 and 4), it should be just after paragraph 3.2, page6 Besides the already specified “Adequate and LGA”. The p values in the column (half Table “According to the main conditions”) are also ambiguous as they seem to test all conditions together (thyroid, immune, renal, other) vs “No risk”. For example, for preeclampsia line the p value is 0.0336, but, if we look at thyroid the percentage is 10.7% vs 9.0% no risk. I would delete this p-value column (and keep the one of the “according to groups” only). But in comments in the result paragraph, I would specify for example that for renal condition and preeclampsia the OR is 5.7 [1.3-21.9], p= 0.002, other OR 2.3 [0.77-6.4], p= 0.06, renal+ other OR 3.0 [1.2-7.3], p= 0.03. For SGA renal OR 5.1 [0.99-21.3], p= 0.04, renal+other OR 3.1 [1.1-8.3], p= 0.02 .

Table 3 and 4. Idem, Some interesting OR could be calculated and written in the results Typing errors. In the legends of Table 3 and 4 and elsewhere in the text “pregnacies” instead of “pregnancy” Line 263, page 11 “outcome: delivery”, should it be “timing of delivery”?

Author Response

Thank you very much for the comments and suggestions aimed at improvoing the quality of our study.

Please find the point by point answers (changes are in red) in the text, and in the following lines.

Comment:

The subject is interesting for teams who have to manage women with egg donations (ED). First, this cohort is impressive: 296 pre or menopaused  women (44 years) in 11 years (ap. 27 per year).

The paper is well written, the introduction, the abstract, the discussion are well conducted with good references. The methodology and calculations are good. The authors explain well the limitations of the study and that, however, they may highlight their findings on kidney function, and that screening of creatinine may be of importance in these women (and put them under low-dose aspirin if pathological for example?).

Answer:

thanks for your remark, we added the following comments on this regard:

Conclusions: according o your co-reviewer's suggesion: This finding stresses the importance of higher awareness of the role of kidney function and of systematic assessment of serum creatinine in all pregnancy (and even more so in pregnancies with potential risk factors, including egg donation).

Furthermore, in the discussion:

Such a policy may also allow preemptive interventions, such as aspirin prophylaxis, and strict monitoring in patients with reduced kidney function.

Comment:

In Tables 1 to 4. I would recommend to skip Adequate and LGA in these 4 tables and keep only the SGA in “size for GA”. First, because SGA is the more important, and, second, because in the p value column, Table 1, we obtain a global 0.02 (“size for GA”), while SGA is 9.1% vs 10.1%, adequate 78% vs 82%. The difference indeed comes from the LGA (!) 12.5% (ED) vs 7.6% ctrls, OR 1.7 [1.2-2.6], p= 0.002.

The Tables give a lot of informations, and there is no place to put the odds ratios in additional columns, however, I would recommend to write down some of them when they are important in the results:

Answer:

We partially agreed with this suggestion: our obstetricians consider that it is important to keep the full description, but we added specific comments, to try to summarize the data:

table 1

The prevalence of small for gestational age babies was apparently not affected by the presence of comorbidity (considered all together), but was twice as frequent after egg donation than in the control population (table 1).

table 2

More in detail, in the subset of women who underwent egg donation, the presence of renal, immunologic or other comorbidity was associated with a higher prevalence of small for gestational age babies, with respect to no risk cases or to individuals with a history of thyroid diseases (table 2).

table 3

In this context, creatinine was more frequently assessed in complicated pregnancies, mainly in the context of preeclampsia and preterm delivery (Table 3). This is also witnessed by the association of creatinine availability with preeclampsia (24.2% vs 1,7%) and with small for gestational babies (12.9% vs 6.4%).

Comment:

In comments of Table 1, paragraph 3.1.

It is important to emphasize first that the preeclampsia rate is of 11.1% in ED women as compared with controls of the whole population (1.7%). OR 7.2 [4.2-12.5], and that they have 3.7% vs 0.5% of premature < 32 weeks OR 7.7 [3.0-20], while, surprisingly, they have the same rate of SGA. This is no more the case when we consider ED’s: morbid vs no-morbid SGA 12.7% vs 6.7% OR 2.01 [0.90-4.6], p= 0.04. Further, it is of note that ED women have significantly more LGA than controls 12.5% (ED) vs 7.6% ctrls, OR 1.7 [1.2-2.6], p= 0.002 (not shown in the Table).

Answer:

thanks for the remark.

we added the following comments, as you suggested:

It is important to emphasize that the preeclampsia rate is of 11.1% after egg donation as compared 1.7% in low risk controls  , and that they have 3.7% vs 0.5% of early preterm babies (< 32 weeks, OR 7.7) while, quite surprisingly, SGA rate differs only when specific comorbidities are considered. Further, it is of note that women who underwent egg donation have a significantly higher incidence of LGA babies (12.5% vs 7.6% in controls).

Comment

Table 2. Table 2 is on page 9 (after Table 3 and 4), it should be just after paragraph 3.2, page6

thanks, we will tag this to the editorial board, since they are in charge of the final formatting. 

Comment

Besides the already specified “Adequate and LGA”. The p values in the column (half Table “According to the main conditions”) are also ambiguous as they seem to test all conditions together (thyroid, immune, renal, other) vs “No risk”.

Indeed, they are Chi square tests for the difference in the overall distribution. We added this information in the legend. 

Comment

For example, for preeclampsia line the p value is 0.0336, but, if we look at thyroid the percentage is 10.7% vs 9.0% no risk. I would delete this p-value column (and keep the one of the “according to groups” only). But in comments in the result paragraph, I would specify for example that for renal condition and preeclampsia the OR is 5.7 [1.3-21.9], p= 0.002, other OR 2.3 [0.77-6.4], p= 0.06, renal+ other OR 3.0 [1.2-7.3], p= 0.03. For SGA renal OR 5.1 [0.99-21.3], p= 0.04, renal+other OR 3.1 [1.1-8.3], p= 0.02 .

Table 3 and 4. Idem, Some interesting OR could be calculated and written in the results.

Answer.

We take the liberty of disagreeing with this comment, since we consider that (also taking into account the usual policy of the JCM), p values are needed, and we prefer relying on the OR of the multivariate analysis, which are less affected by the high heterogeneity of the population.

Following the suggestion of the reviewer, we added some OR also in the text.

Typing errors. In the legends of Table 3 and 4 and elsewhere in the text “pregnacies” instead of “pregnancy” Line 263, page 11 “outcome: delivery”, should it be “timing of delivery”?

thanks, we corrected them.